# Emerging Chemotherapy Targets: Insights from Advances in Glioma Treatment

**DOI:** 10.3390/biomedicines13061452

**Published:** 2025-06-12

**Authors:** Rogina Rezk, Abanob George Hanna, Hunter Hutchinson, Mariam Farag, Brandon Lucke-Wold

**Affiliations:** 1College of Medicine, University of Florida, Gainesville, FL 32610, USA; roginarezk@ufl.edu (R.R.); ahanna3@ufl.edu (A.G.H.); hutchinsonhunter@ufl.edu (H.H.); mariamfarag@ufl.edu (M.F.); 2Department of Neurosurgery, University of Florida, Gainesville, FL 32608, USA

**Keywords:** primary brain tumors, glioma, glioblastoma, tumor blood–brain barrier, immunosuppression, blood–brain barrier

## Abstract

Primary brain tumors represent a significant focus of contemporary research. With advancements in technology and the increasing detection of cases through novel diagnostic methods, innovative therapies and approaches to chemotherapy continue to emerge. The paper explores recent advancements in chemotherapy for glioblastoma, highlighting innovative approaches that provide valuable mechanistic insights. It delves into the mechanisms of action, molecular targets, and the future potential of emerging therapies for gliomas. Additionally, this paper offers an overview investigating a range of therapies, including various chemotherapeutic agents, CAR-T cell therapies, drugs targeting cellular respiration, and other approaches. Furthermore, the paper addresses chemotherapy-related challenges, including the blood–brain barrier, drug resistance, and immunosuppression, while proposing potential solutions to overcome these obstacles.

## 1. Introduction

Gliomas are primary brain tumors derived from neuroglial cells. Primary brain malignancies are cancers that originate within the brain tissue itself, with the potential to spread to adjacent brain structures and nerves. Depending on the malignancy’s nature, these tumors can be either cancerous or benign, but both types can cause significant symptoms that may be easily overlooked. Common symptoms include headaches, speech difficulties, vomiting, seizures, and other nonspecific manifestations that can complicate early diagnosis [1]. The treatment options for brain tumors depend on the grade, stage, and prognosis of the tumor. Some of these treatment options include surgery, radiation therapy, radiosurgery, chemotherapy, and targeted therapy [2].

Chemotherapy is a critical component in the treatment of primary brain tumors, particularly following surgical intervention. For instance, temozolomide, a chemotherapy drug approved by the FDA in 2013, is utilized as a potential adjuvant treatment for Glioblastoma Multiforme (GBM) [3], the most prevalent and aggressive glioma in adults. Temozolomide plays a vital role in enhancing the effects of surgery and radiation therapy, aiming to eradicate any remaining tumor cells and prevent metastasis [4]. Furthermore, chemotherapy with temozolomide significantly reduces the likelihood of tumor recurrence, improving patient outcomes and survival rates [5,6]. Even when a primary brain tumor is not curable, chemotherapy remains essential for palliative care. Palliative care helps manage symptoms, improve quality of life, and potentially extend patient survival by controlling tumor growth and reducing neurological deficits [7].

Chemotherapy is an evolving field, driven by advancements in our understanding of tumor biology and technological innovations that enhance molecular diagnostics and early detection. These developments contribute to improved prognoses for patients. This review article aims to highlight some of the significant advancements and emerging innovations in chemotherapy that are shaping the future of glioma treatments. Thus, our review includes many examples of targets in glioma.

## 2. Current Challenges in Chemotherapy for Gliomas

### 2.1. Blood–Brain Barrier

The blood–brain barrier (BBB) is a selectively permeable barrier that separates the blood from the brain’s interstitial fluid. It tightly controls the movement of molecules and ions from blood vessels into the brain, ensuring regulated access to brain tissue [8]. Tight Junctions between endothelial cells (ECs) prevent the passive diffusion of large, polar molecules across the barrier. ECs of the BBB differ from ECs in other vasculature as they have a more comprehensive network of tight junctions and limited trans-endothelial pathways; both characteristics severely limit the movement of molecules across the membrane [9]. Astrocytes are responsible for maintaining the tight junctions, while pericytes surrounding the endothelium have long processes that contain contractile proteins that regulate the capillary diameter [10].

Because of its structure, the BBB was long thought to be a limiting factor in delivering chemotherapy to primary or metastatic brain tumors since many of the drug agents are not able to cross the BBB due to their size and polarity. However, new knowledge of BBB alterations in brain tumors has modified this previous notion. The alterations to the integrity of the BBB in primary/metastatic brain tumors are often called the tumor blood–brain barrier [11]. This indicates that the primary distinction between the tumor brain barrier and the normal blood–brain barrier lies in its structural abnormalities, particularly increased fenestration, which is commonly driven by the overexpression of vascular endothelial growth factor (VEGF). Increased expression of VEGF leads to reduced expression of tight junctions, contributing to increased paracellular diffusion and vascular leakage [12]. In addition, overexpression of VEGF enhances permeability by promoting the formation of transcellular transport vesicles [13]. This information may be valuable for drug delivery applications, as preclinical studies have demonstrated that exogenous administration of VEGF increases BBB permeability, thereby enhancing the efficacy of nanomedicine delivery, including liposomal doxorubicin [14].

However, the structure of the tumor blood–brain barrier is heterogeneous. At the edge of the tumor, the BBB remains intact. As the transition from the edge to the core of the tumor, the tumor blood–brain barrier becomes severely altered with increased permeability [15]. In other words, unlike the VEGF-driven increase in permeability at the core of the blood–tumor barrier, medication distribution to the periphery of the tumor is severely limited since these areas usually have an intact or only slightly compromised BBB. Treatment resistance and poor clinical outcomes are all influenced by this unequal permeability throughout the tumor, which results in insufficient drug penetration into infiltrative tumor cells at the periphery [16,17].

In the case of glioblastoma (GBM), emergent studies use new techniques, such as dynamic contrast-enhanced magnetic resonance (DCE-MR), to quantify the transport of contrast molecules [18]. This method allows to have a better understanding of the intactness of the BBB in GBM. The results of these studies have shown that there is great intra-tumoral heterogeneity [18]. In other words, GBM can have both intact BBB spots and disrupted BBB spots within the same tumor. This variability poses a question of how successful chemotherapy can be. In essence, chemotherapeutic drugs can penetrate portions of the tumor in which the BBB is disrupted, but they would not be effective against the entire tumor.

### 2.2. Molecular Resistance

One of the challenges of chemotherapy is the development of chemoresistance in tumors. Glioblastoma (GBM) is a good model for this concept. This is due to its heterogeneity. Distinct subtypes of GBM can coexist within the same tumor, and over time, cancer stem cells (CSCs), which are capable of differentiation, can accumulate successive mutations, promoting resistance [19]. Gene expression profiling showed 418 genes upregulated in tumor tissues vs. CSCs and 44 different genes upregulated in CSCs vs. tumor tissue [20]. This increased heterogeneity is associated with worse patient survival [21]. For example, this heterogeneity reduces the efficacy of CAR-T therapy as they are designed against specific tumor antigens; therefore, the efficacy of this new therapeutic approach is often blunted by antigenic variation by tumor cells [22]. To overcome this limitation, newer CAR-T therapies aim to create chimeric receptors with multiple antigen recognition sites [22].

Additionally, there are molecular resistance mechanisms that play a role post-transcriptionally, including microRNAs (miRNAs). These are 18–22 nucleotide-long non-coding RNA segments that affect gene expression [23]. The mechanism by which these miRNAs play a role in the progression of the disease and its resistance varies even among the different types of malignancies. miRNA-566 was found to be overexpressed in GBM [24]. By inhibiting this miRNA, invasion and angiogenesis of GBM were suppressed, suggesting an oncogenic role of this miRNA [24]. It is noteworthy that the biogenesis of miRNA itself is often regulated by oncogenic proteins, including p53 [25]. Another non-coding RNA that contributes to the molecular resistance of GBM is long non-coding RNAs (lncRNAs). For example, lncRNA SLC26A4-AS1 is downregulated in GBM [26]. lncRNA SLC26A4-AS1 attracts NFKB1 to the NPTX1 promoter, leading to its upregulation. Increasing the activity of NPTX1 is tumor suppressive, and therefore, the downregulation of lncRNA SLC26A4-AS1 causes loss of tumor suppression and oncogenesis [26].

Another molecular mechanism of resistance is redundancy [27]. In GBM, cells often have different signaling pathways to achieve the same result. One striking example is angiogenesis. Vascular endothelial growth factor (VEGF) remains the key stimulator of angiogenesis. However, there are several factors—FGF, PDGF, HGF, agnoproteins, and interleukins—that promote angiogenesis [28]. GBM can utilize a number of these factors to circumvent the inhibition of anti-VEGF therapies. In even more aggressive tumors, cells can completely bypass these mechanisms by vascular mimicry (VM) [26,27]. VM is the formation of blood vessel-like structures that do not have any endothelium [26]. These vessel-like structures are enough to support the growth of the tumor. As a matter of fact, tumor cells begin to express endothelial markers, including CD31, VE-cadherins, and Factor VIII [29]. Unfortunately, using standard antiangiogenic therapies has shown disappointing results against tumors exhibiting VM.

### 2.3. Immunomodulation

Immunomodulation of the tumor microenvironment helps cancer cells evade the immune system by the mechanisms summarized in Figure 1. For example, cancer cells often downregulate MHC presentation on their surface, reducing the presentation of neoantigen and evading recognition by immune cells [30]. In GBM, both MHC I and MHC II are downregulated. This causes a broader effect on the immune system. Another mechanism of immunosuppression utilized by primary brain malignancies is overexpression of programmed death-ligand 1 (PD-L1) [31]. PD-L1 on cells activates T-regulatory cells and induces T-cell apoptosis [32]. Overall, the increased interaction between PD-L1 and PD-1 induces T-cell exhaustion and anergy, reducing their ability to effectively kill tumor cells.

Tumor growth factor β (TGF-β), interleukin-10 (IL-10), and prostaglandin E2 are rich in the microenvironment of tumor cells [30]. All three of which have immunosuppressive effects. IL-10 is mainly secreted from tumor-associated pro- and anti-inflammatory microglia and macrophages (TAMs) [33]. TAMs are thought to make up 50% of the tumor mass in GBM [33]. IL-10 in combination with IL-4 (also expressed in GBM) can polarize macrophages to be more M2-like, exhibiting anti-inflammatory properties [33]. As a matter of fact, in a study by Bloch and his colleagues, it was demonstrated that macrophages in gliomas have increased expression of B7-H1, which induces T cell apoptosis [34]. Additionally, TGF-β, which is found to upregulate claudin-4 (CLDN 4) CLDN 4 was found to allow tumor cells to gain migratory, infiltrative properties [35].

A new area of research interest is the role of epigenetics in immunomodulating the tumor microenvironment (TME) of GBM. One example of this was explored by Gangoso and his colleagues [36]. In their study, it was noted that methylation was significantly reduced in the Irf8 promoter region of triple mutant cell lines, resembling the most aggressive form of GBM [36]. The Irf8 has a significant effect on immune evasion, and it is activated by IFN-gamma from macrophages [36]. Reduced methylation of the interferon regulatory factor 8 (Irf8) promoter results in increased Irf8 expression in aggressive forms of glioblastoma multiforme (GBM). By pushing myeloid-derived suppressor cells (MDSCs) toward immunosuppressive states, this change helps the tumor avoid immune detection and plays a crucial part in reshaping the tumor immune microenvironment [37]. In preclinical GBM models, therapeutic approaches that boost Irf8 activity, such as gene therapy utilizing retroviral replicating vectors, have shown decreased immunosuppression and decreased tumor development. However, the varied permeability of the blood–tumor barrier (BTB) and blood–brain barrier (BBB) limits the efficacy of IRF8-targeted treatments [37]. Other studies show a positive correlation between the number of macrophages in the TME and decreased survival, given the same grade of tumor [34]. An increased number of macrophages is associated with immunosuppression, which in turn promotes tumor development, invasion, angiogenesis, and resistance to therapy, ultimately leading to poor outcomes [38]. Interestingly, epigenetics can also be used to overcome modulation. Lofiego and his colleagues demonstrated that guadecitabine can hypomethylate promoter regions that upregulate certain genes associated with the activation of T and B cells, as well as increase MHC II presentation [39]. This epigenetic remodeling enhances antigen processing and presentation, processes typically suppressed in GBM to evade the immune system, thereby shifting GBM cells toward a more immunogenic phenotype and improving therapeutic efficacy [39]. Other pre-clinical studies have demonstrated strategies to modify the tumor microenvironment using targeted gene therapy, such as engineering olfactory ensheathing cells to express HSV thymidine kinase (OEC-TK) [40]. The primary mechanism involves delivering suicide genes directly to the tumor microenvironment (TME) to induce selective tumor cell death.

### 2.4. Toxicity

Current chemotherapies are associated with many toxicities. Temozolomide is used in the treatment of anaplastic astrocytoma and GBM [41]. Nausea, vomiting, anorexia, thrombocytopenia, anemia, and AST/ALT increases are all common toxicities associated with this drug [42]. Proteinuria is the second most common toxicity of bevacizumab, and it is also a dose-dependent toxicity [42]. Grade 3 proteinuria incidence increased from 0.0 to 0.1% in the control arm to 0.8–4.0% in the therapy arm in a randomized control trial [42].

## 3. Emerging Targets in Glioma Chemotherapy

Treatment of primary brain malignancies is an area of active research to manage these patients better. One area of emerging treatment is targeted therapies. These therapies are developed to target pathways that promote cancer growth. Tyrosine kinases (TK) are one of these molecular targets. TKs activate tyrosine residues of the receptor and intracellular proteins, activating a cascade of second messengers [43].

Epidermal Growth Factor Receptor (EGFR) TK is often mutated in primary brain tumors, including GBM. It promotes cell proliferation and invasiveness. Gefitinib is a drug that target EGFR and is involved in clinical trials to determine its efficacy [44]. While gefitinib has demonstrated limited efficacy in some studies, as shown in Table 1, its inability to penetrate the blood–brain barrier (BBB) prevents it from being a standard treatment for glioblastoma (GBM). However, its observed efficacy in early-phase trials remains valuable for guiding the development and optimization of targeted therapies that exploit similar pathways. It is also noteworthy that its benefits have not been substantial in the broader patient population [44].

Platelet-derived growth factor (PDGFR) is another TK that has been a center of attention; it, too, is overexpressed in primary brain cancer, promoting tumor growth. However, imatinib, a tyrosine kinase inhibitor, showed no significant benefit in phase II clinical trials [45]. The inefficacy of imatinib in treating GBM stems from the intricate nature of the tumor and its microenvironment and some of the limitations of the drug, notably its restricted ability to penetrate the blood–brain barrier [45]. Nonetheless, emerging studies instead aim to inhibit molecules downstream of PDGFR. One of these is SHP2 phosphatase, a nonreceptor protein tyrosine phosphatase encoded by the PTPN11 gene [46]. SHP-2 mediates the signaling of PDGFR to promote its effect [46]. Inhibiting SHP-2 might prove clinical utility. A study particularly examined SHP099, an inhibitor of SHP-2, in mouse models [47]. In cell lines, it was shown that SHP099 specifically blocks ERK1/2 phosphorylation and proliferation of cells with overexpressed RTK, including PDGFR [47]. Not only that, but in vivo experimentation with mice has also shown that SHP099 reaches efficacious concentrations in the brain and improves survival rates in GBM xenograft-bearing animals alone and with TMZ [47]. There are currently no clinical trials on SHP-2 inhibitors, but basic science research shows great promise.

Additionally, there are emerging antibody-based therapies. Bevacizumab is a monoclonal antibody that targets vascular endothelial growth factor (VEGF). As mentioned earlier, VEGF is overexpressed in the tumor environment, promoting angiogenesis. Bevacizumab is thought to halt this pathological process [48]. While acknowledging bevacizumab’s potential efficacy, it is critical to consider its limited impact on overall survival compared to its demonstrated effectiveness in improving progression-free survival [49,50].

Another promising avenue of brain cancer treatment research is immunomodulation. Chimeric antigen receptor T-cells (CAR-T cells) are engineered T cells with antigen receptors specific to neoantigens presented by tumor cells [51]. This approach involves modifying T-cells to target specific tumor antigens, enhancing their proliferation and function to fight the tumor [51]. Although CAR-T is one of the most effective adoptive cell therapies, one of the issues with it is that the specificity of CAR-T cells makes it difficult to target heterogeneous tumors [46]. While the mechanisms by which CAR-T cells operate are complex and may present limitations, multiple clinical studies, such as the study by Brown et al. [52], have demonstrated the efficacy of CAR-T cell therapy in treating progressive GBM (as shown in Table 1). The mitochondria have also been a target for certain chemotherapies. Pyruvate dehydrogenase alpha (PDHA) inhibitors (currently in phase III trials), such as CPI-613 (devimistat generically), work by inhibiting the enzymes pyruvate dehydrogenase and α-ketoglutarate dehydrogenase (OGDH) specifically in tumor cells [53].

As an analog of lipoic acid, devimistat disrupts the Krebs cycle, leading to metabolic dysfunction and selective cell death in the tumor [54]. Another therapeutic drug that targets mitochondria is gamitrinib. However, instead of halting the TCA cycle, gamitrinib inhibits the function of complex II of the respiratory chain by antagonizing mitochondrial heat shock protein 90 (Hsp90) as well as by targeting TNF receptor-associated protein 1 (TRAP1) to induce cell death via disrupting mitochondrial respiration in cancer cells [54]. Gamitrinib has shown efficacy in GBM rat models when combined with BH-3 mimetics. BH-3 mimetics induce the release of BAX and BAK from BCL-2 and Bcl-xL, which in turn induce apoptosis. There is a current Phase I clinical trial investigating gamitrinib for advanced, non-brain, cancers [54], which could help us understand its mechanism of action to better target other types of malignancies, with the results summarized in Table 1. Other drugs, such as vorasidenib, are also a recently FDA-approved drug that targets mitochondrial isocitrate dehydrogenase (IDH) mutations, as shown in Table 1. Vorasidenib (AG-881) is an oral medication that can enter the brain and inhibit both IDH1 and IDH2 enzymes, commonly in low-grade gliomas [55]. Through this mechanism, vorasidenib prevents the buildup of d-2-hydroxyglutarate (2-HG), a substance that plays a critical role in cancer development [56]. Currently, the drug is being tested in clinical trials to treat cancers that involve IDH mutations, with the findings of a study by Mellinghoff et al. [56], outlined in Table 1.

In addition, ongoing research on BMX-01 (BMX-HGG), a metalloporphyrin, has demonstrated its potential as a novel therapy when used concurrently with radiation. BMX-001 (MnTnBuOE-2-PyP^5+^) is a manganese-based compound that acts similarly to the enzyme superoxide dismutase (SOD) [57]. It helps reduce oxidative stress by converting superoxide ions (O_2_^•−^) into hydrogen peroxide (H_2_O_2_). In addition to this SOD-like function, BMX-001 has other redox properties that allow it to engage with various molecules within the cell, supporting its broader protective actions [57].

Due to its mechanism of reducing oxidative stress, BMX-001 has demonstrated the potential to selectively protect healthy cells from radiation-induced damage while simultaneously enhancing the radiosensitivity of cancer cells. Following the successful completion of Phase I trials, which established its safety profile, BMX-01 is now undergoing Phase II trials to further evaluate its efficacy, as shown in Table 1.

While some of these mentioned drugs have shown promising effects, as highlighted in Table 1, it is crucial to emphasize that despite current advancements and promising therapies under investigation, gliomas remain an aggressive and challenging cancer with no curative treatments available. While certain therapies have shown potential in preclinical and early clinical trials, as shown throughout the paper and in Table 1, the prognosis for glioma patients remains poor. Ongoing research is essential to explore new therapeutic options that might eventually offer more effective treatments and improved outcomes.

**Table 1 biomedicines-13-01452-t001:** Overview of study results for the specified drug agents.

Study ID	Drug	Results
29492119	Gefitinib	Gefitinib significantly improved outcomes in patients with the EGFR mutation and wild-type PTEN, achieving a progression-free survival (PFS) of approximately 9 months and an overall survival (OS) of about 20 months. Gefitinib demonstrated poorer outcomes in GBM patients with concurrent EGFR and PTEN mutations, with PFS and OS of 6 months and 9 months, respectively [58].
31119479	Imatinib	The median progression-free survival (PFS) with imatinib was approximately 4 months, and the overall survival (OS) was about 16.6 months, compared to 8 months in the control group. Patients with the 1p/19q codeletion demonstrated a higher OS of 19.2 months. Of the participants, 61% experienced adverse effects, including fatigue, hemorrhage, gastrointestinal issues, and hypophosphatemia. These findings warrant validation in a larger cohort [59].
30115593	Bevacizumab	Between 2011 and 2015, 155 patients received either monotherapy of temozolomide (n = 77) or combination therapy of temozolomide and bevacizumab (n = 78). At 12 months, overall survival was 61% for monotherapy and 55% for combination therapy. Grade 3 or 4 hematological toxicity was higher in the combination group (33% vs. 23%). Common adverse events included nervous system disorders, fatigue, and nausea, all more frequent with combination therapy. Infections were also higher in the combination group (38% vs. 23%), with one treatment-related death reported [60].
37316802	Bevacizumab	Bevacizumab (BEV) improves progression-free survival, palliation, and cognitive function in recurrent glioblastoma (rGBM), but overall survival benefits lack strong evidence. BEV combined with lomustine, or radiotherapy is more effective than monotherapy. Better responses are predicted by IDH mutation, large tumor burden, and double-positive signs. Low-dose BEV is as effective as the standard dose, but optimal timing remains unclear. Further studies are needed [61].
28426845	CAR-T Cells	Seventeen patients with progressive HER2-positive glioblastoma (10 adults, 7 children) received autologous HER2-CAR VST infusions without prior lymphodepletion. Infusions were well tolerated, with no dose-limiting toxic effects, and HER2-CAR VSTs were detectable in peripheral blood for up to 12 months. Of 16 evaluable patients, 1 had a partial response for over 9 months, 7 had stable disease (8 weeks to 29 months), and 8 progressed. Three patients with stable disease remained progression-free for 24–29 months. Table 1: Overview of Study Results for the Specified Drug Agents. The median overall survival for the cohort was 11.1 months from the first infusion and 24.5 months from diagnosis. Phase 2b study is warranted [51].
260959190	CAR-T Cells	The study demonstrates the feasibility of producing autologous CTL clones expressing an IL13 (E13Y)-zetakine CAR for targeted HLA-independent glioma treatment. Intracranial infusion of these CTLs into three patients with recurrent glioblastoma was well-tolerated, with manageable brain inflammation. Two patients showed transient anti-glioma responses, and one patient’s tumor tissue analysis revealed reduced IL13Rα2 expression post-treatment. MRI of another patient showed increased tumor necrosis at the infusion site [52].
28724573	CAR-T Cells	The infusion of EGFRvIII-directed CAR T cells in recurrent glioblastoma (GBM) patients was well-tolerated and showed temporary T cell expansion in the blood. Post-treatment tumor analysis revealed antigen loss in five patients, but adaptive resistance mechanisms emerged, including upregulation of inhibitory molecules and increased regulatory T cell infiltration [62].
35129069	Gamitrinib	The study showed that Gamitrinib exhibits significant anticancer activity in various cancer cell lines, including colon adenocarcinoma, breast adenocarcinoma, and melanoma. It effectively inhibits cancer cell growth, both as a monotherapy and in combination with other therapies, with favorable pharmacokinetics and minimal toxicity in preclinical models (rats and beagle dogs). Toxicity studies found no major adverse effects on the tested doses [54].
37272516	Vorasidenib	In a study of 331 patients, 168 received vorasidenib and 163 received a placebo. After a median follow-up of 14.2 months, progression-free survival was significantly longer in the vorasidenib group (27.7 months) compared to the placebo group (11.1 months). Grade 3 or higher adverse events occurred in 22.8% of patients receiving vorasidenib, compared to 13.5% in the placebo group [56].
PMC9354202	BMX-01 (BMX-HGG)	Fifteen patients (ages 19–80) with WHO grade 4 glioblastomas underwent neurocognitive testing before and after radiation therapy (RT). Most had neurocognitive impairment, with deficits ranging from 46.7% to 80% on specific tests. However, at two months (n = 15) and six months (n = 9) after treatment, most patients showed improved neurocognitive performance. Neurocognitive function can be maintained or improved in patients receiving concurrent radiation therapy and temozolomide, along with BMX-001 treatment [63].

## 4. Overcoming Possible Challenges

Achieving effective chemotherapeutic delivery for gliomas involves overcoming various anatomical and tumor-specific hurdles. The initial challenge lies in navigating the blood–brain barrier (BBB), essential for chemotherapy drug access. Subsequently, addressing the diversity and drug resistance of tumors poses further complexities in targeted therapy selection. Moreover, systemic challenges include treatment costs, long-term survivorship, side effect management, and issues with clinical trial recruitment. Alongside these hurdles, there is a critical need to advance personalized medical approaches for gliomas.

### 4.1. BBB and Physiological Barriers

Chemotherapy remains a cornerstone in the treatment of primary brain tumors. However, as previously mentioned, its efficacy is often limited by various challenges, particularly the anatomical blood–brain barrier (BBB). To enhance therapeutic delivery, strategies such as intrathecal chemotherapy, focused ultrasound with microbubbles, and nanoparticles capable of crossing the BBB can be employed, as summarized in Figure 2.

Intrathecal chemotherapy has been used for decades to deliver medication via CSF. While it bypasses the BBB, barriers in the subarachnoid space and high CSF turnover still limit the drug concentration reaching tissue targets [64]. Hydrophilic drugs are cleared quickly as the CSF clears, hydrophobic drugs have limited solubility and remain mostly at the injection site, and larger molecule medications have a lower likelihood of crossing the subarachnoid barriers. Thus, although intrathecal chemotherapy administration provides an alternative route of drug delivery, there remains a need for more research on the better delivery of drugs within CSF [65].

Another means of overcoming the BBB is by using focused ultrasound and microbubbles to enhance delivery. Using focused delivery of ultrasound with microbubbles to direct the exact location of drug delivery has been researched as a way of disrupting the BBB and injecting drugs directly [66]. The use of ultrasound adds energy that results in oscillation of the microbubbles, which then disrupts the BBB and allows for the targeted delivery of drugs at the site of ultrasound activation. In a preclinical study by Chen et al., focused ultrasound-induced BBB opening facilitated the delivery of interleukin-12 for brain tumor immunotherapy, demonstrating enhanced drug delivery and antitumor effects [67].

In addition to the use of focused ultrasound and microbubbles, nanoparticles have shown promising outcomes for the delivery of drugs directly to primary brain tumors through various routes. An article by Hersh et al. showed that developed nanoparticles capable of crossing the blood–brain barrier (BBB) and successfully delivering drugs for glioma treatment in vivo mouse models [68]. In human subjects, exosome membrane-coated nanoparticles have also proven effective in tumor treatment and regression [69]. Additionally, biocompatible magnetic iron oxide nanoparticles have demonstrated success in delivering doxorubicin to primary brain tumors. This approach holds the potential for overcoming multidrug resistance by ensuring targeted drug delivery [70]. The use of biodegradable wafers, which are wireless devices that are placed near a tumor after surgery, has been used as a method to deliver cancer drugs. Although there is a promising technique, more research needs to be completed to determine suitable material and to determine the prognosis associated with the use of wafers [67]. These advancements highlight the potential of nanoparticles in enhancing chemotherapeutic efficacy against glioblastoma through innovative delivery mechanisms.

### 4.2. Intrinsic Tumor Barriers

As previously mentioned, gliomas exhibit significant heterogeneity, with different regions of the tumor harboring diverse genetic and phenotypic profiles [71]. This complexity makes it difficult for a single therapeutic approach to effectively target all tumor cells. The heterogeneity of tumors can be combated with the use of combination drug therapies. However, targeted drug delivery to the brain is complicated by various barriers. In addition to combination drugs, tailoring treatments based on tumor markers and mutations is a promising path for better management of gliomas. A study by Patel et al. demonstrated the effectiveness of personalized treatment plans for glioblastoma, where genetic profiling informed the use of specific targeted therapies, resulting in improved patient outcomes [72]. Personalized treatment and genetic profiling of tumors hold promise for overcoming the challenges posed by tumor heterogeneity and the blood–brain barrier, ultimately leading to more effective treatments for glioblastoma and other primary brain tumors.

As briefly mentioned, another challenge in finding and developing chemotherapeutic targets is resistance to certain drugs. Nanomedicine is one possible solution as it expands the chemotherapeutic drugs that can be delivered to the brain. A study found that nanoparticle-based delivery systems could overcome drug resistance in cancer therapy. This study showed that nanoparticles loaded with chemotherapeutic drugs successfully bypassed efflux pumps and increased drug concentration within resistant tumor cells [73]. Another approach to decreasing the incidence of resistance and relapse is by targeting the tumor cell environment, such as stromal cells, immune cells, and other cells that aid in the growth of the tumor. This strategy can be helpful in tumor regression treatment and in preventing metastasis to distant sites. In a research paper on microenvironment regulation of metastasis, Joyce and Pollard concluded that further research targeting the stromal cells of tumors could provide methods for halting tumor growth and metastasis [74].

### 4.3. Systemic Challenges

There are many challenges in developing tumor treatments beyond drug delivery and research on tumor biology and metastasis. The development of drugs and technologies for drug delivery is significantly expensive. The methods of drug delivery and drug targeting are also costly for both the healthcare system and patients. Further research into creating generic drugs is needed, along with policy reforms to ensure equitable access to these drugs [75,76]. A study reviewing the economic burden of glioblastoma and the cost-effectiveness of pharmacologic treatments, such as temozolomide and carmustine wafers, found that while these treatments may offer better clinical benefits, their high costs present a challenge for cost-effectiveness [76]. Although both temozolomide and carmustine wafers improve survival rates, determining if the improved survival justifies the increased cost is what determines the cost-effectiveness ratio, which determines which therapies are offered to patients. Thus, the cost of treating gliomas, especially glioblastomas, raises ethical questions for both patients and the healthcare system.

### 4.4. Prognosis and Long-Term Survivorship

There are significant challenges with long-term survivorship and quality of life in patients with glioblastomas. A systematic review of the 10-year survival rates of glioblastomas concluded that the rates were as low as 2.3% [77]. In addition to low 5- and 10-year survival rates, the prognosis and quality of life for patients with glioma are severely impacted. Various studies have measured the impact on quality of life through metrics including physical, emotional, and cognitive functioning, with physical symptoms such as fatigue, pain, seizures, insomnia, cognitive dysfunction, and sleep dysfunction being the most reported [78]. The poor survival rates and prognosis further complicate efforts to find effective treatments and improve patients’ quality of life.

## 5. Conclusions

Non-specific chemotherapy, used in conjunction with surgical resection and radiotherapy, is the standard of care for gliomas. Challenges in treating gliomas include toxicity, molecular resistance, and difficulty penetrating the BBB. Cancer cells modulate the tumor microenvironment to be immunosuppressive through mutations such as MHC downregulation and PD-L1 upregulation [31,32]. Brain tumors also modify the BBB’s permeability in a heterogeneous fashion, making consideration of the brain–tumor barrier important for future chemotherapies [15,18].

Novel chemotherapies for gliomas target specific cancer growth pathways. Imatinib, which inhibits the tyrosine kinase platelet-derived growth factor receptor, struggles to pass the BBB, leading it to be ineffective for GBM in phase II clinical trials [45]. Bevacizumab targets vascular endothelial growth factor and has potential efficacy in primary CNS tumors, but with marginal improvements in overall survival [48,60,61]. Intracranial administration of CAR-T cells provides bespoke treatment of tumor antigens and shows promise in GBM.

The efficacy of glioma chemotherapies can be improved with better delivery methods. Drugs can be designed to be better distributed when administered intrathecally [65]. Microbubbles can transport chemotherapeutic particles across a BBB disrupted with focused ultrasound [66]. Nanoparticles and biodegradable wafers can also achieve targeted chemotherapy delivery [67].

Further development and clinical trials may lead to mutation-targeting chemotherapies and improved delivery methods, which lead to a better prognosis for patients with gliomas. An effective treatment for gliomas must be specific to tumor cells, liposoluble to cross the BBB, and achieve therapeutic concentration without serious adverse effects. On top of clinical efficacy, the cost-effectiveness of novel therapeutics must also be considered. In the meantime, patient-care teams need to treat the harsh side effects of current chemotherapy regimens with proactive medical, physical, and mental health therapy.

## Figures and Tables

**Figure 1 biomedicines-13-01452-f001:**
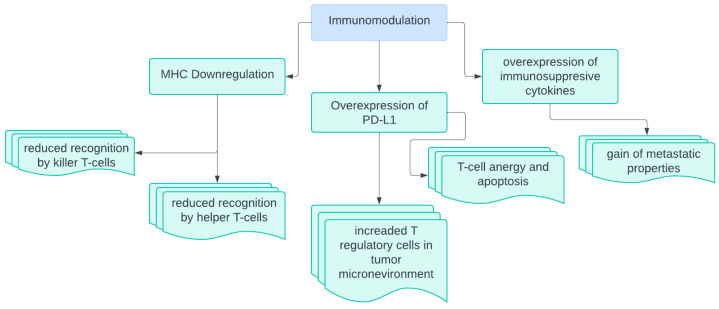
Overview of general mechanisms of immunomodulation.

**Figure 2 biomedicines-13-01452-f002:**
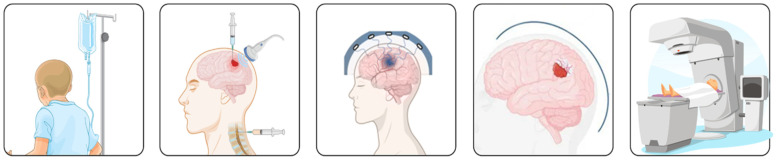
Different types of drug delivery for brain cancers include intrathecal drug delivery, focused ultrasound to disrupt the brain barrier, biodegradable wafers, and nanomedicine delivery.

## Data Availability

Data supporting these findings are available within the article.

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
