# Peer review of "Emerging Chemotherapy Targets: Insights from Advances in Glioma Treatment"

_biomedicines, 2025, doi:10.3390/biomedicines13061452_

Round 1
Reviewer 1 Report
Comments and Suggestions for Authors
Comments for the Authors
In this review article, the authors have explored an intriguing subject that has been somewhat overlooked in recent years, Emerging Chemotherapy Targets: Insights from Advances in Glioma Treatment. Their focus on the recent advancements in this area offers valuable insights into Dementia and more. Considering the limited availability of such articles, this manuscript holds considerable potential for publication in Biomedicines. However, further polishing is required to meet the journal's standards. With its concise content, this article promises to offer valuable insights to scientists. After undergoing no revisions, I endorse the publication of this manuscript in Biomedicines, where it can contribute significantly to the field.
- The manuscript was well written and but can improve the English further.
- Line 68, authors should comment on it. what does the overexpression of VEGF contribute to the disruption of the blood-brain barrier in brain tumors?
- Followup to the above one, are there the implications for drug delivery?
- I am a bit confused with this words mismatch, howere is the term “tumor blood-brain barrier” used instead of just “blood-brain barrier” when referring to brain tumors, and are there the key structural and functional differences between them? Line 68, authors should comment on it.
- Line 75, authors should comment on it. Considering the heterogeneous nature of the tumor blood-brain barrier, what might this influence the design and effectiveness of chemotherapeutic strategies for brain tumors?
- Line 153, authors should comment on it. what does hypomethylation of the Irf8 promoter contribute to immune evasion in aggressive GBM subtypes?
- Followup to the above oen, is there any role of IFN-γ in this context?
- Line 155, authors should comment on it. how is the significance of the observed correlation between macrophage density in the tumor microenvironment and patient survival in GBM?
- Line 158, authors should comment on it. what might epigenetic drugs like guadecitabine be used to reverse immune suppression in GBM?
- Followup to the above one, how the mechanisms underlie their immunostimulatory effects?
Comments for the Authors
In this review article, the authors have explored an intriguing subject that has been somewhat overlooked in recent years, Emerging Chemotherapy Targets: Insights from Advances in Glioma Treatment. Their focus on the recent advancements in this area offers valuable insights into Dementia and more. Considering the limited availability of such articles, this manuscript holds considerable potential for publication in Biomedicines. However, further polishing is required to meet the journal's standards. With its concise content, this article promises to offer valuable insights to scientists. After undergoing no revisions, I endorse the publication of this manuscript in Biomedicines, where it can contribute significantly to the field.
- The manuscript was well written and but can improve the English further.
- Line 68, authors should comment on it. what does the overexpression of VEGF contribute to the disruption of the blood-brain barrier in brain tumors?
- Followup to the above one, are there the implications for drug delivery?
- I am a bit confused with this words mismatch, howere is the term “tumor blood-brain barrier” used instead of just “blood-brain barrier” when referring to brain tumors, and are there the key structural and functional differences between them? Line 68, authors should comment on it.
- Line 75, authors should comment on it. Considering the heterogeneous nature of the tumor blood-brain barrier, what might this influence the design and effectiveness of chemotherapeutic strategies for brain tumors?
- Line 153, authors should comment on it. what does hypomethylation of the Irf8 promoter contribute to immune evasion in aggressive GBM subtypes?
- Followup to the above oen, is there any role of IFN-γ in this context?
- Line 155, authors should comment on it. how is the significance of the observed correlation between macrophage density in the tumor microenvironment and patient survival in GBM?
- Line 158, authors should comment on it. what might epigenetic drugs like guadecitabine be used to reverse immune suppression in GBM?
- Followup to the above one, how the mechanisms underlie their immunostimulatory effects?
Author Response
**all changes are shown in red**
1- A grammatical review was done
2-3) Thank you for your comment. We addressed the role of VEGF overexpression in disrupting the blood–brain barrier (BBB) in lines 72–74. Specifically, we noted that VEGF contributes to BBB disruption primarily through the breakdown of tight junctions and the promotion of transcellular transport vesicle formation, both of which increase vascular permeability in brain tumors.
Furthermore, in lines 77–81, we provided an example of how this altered permeability is being leveraged in current research to improve drug delivery strategies.
4- Thank you for your thoughtful comment. To clarify the terminology, we defined the blood–brain barrier (BBB) in lines 57–58 as “a selectively permeable barrier that separates the blood from the brain’s interstitial fluid.”
In lines 70–72, we introduced the term tumor blood–brain barrier (TBBB) to describe the structural and functional alterations that occur in the BBB in the context of primary or metastatic brain tumors. As noted, this term reflects the compromised and heterogeneous nature of the barrier in tumor regions. Further elaboration in lines 72–75 highlights the structural differences—such as disrupted tight junctions, abnormal vasculature, and altered permeability—that distinguish the TBBB from the intact BBB.
5- Thank you for your comment. We have addressed the heterogeneous nature of the tumor blood–brain barrier (BBB) in lines 85–90, elaborating on how this variability affects drug distribution and therapeutic efficacy. Specifically, we discussed how differences in BBB permeability across tumor regions can influence the effectiveness of chemotherapeutic agents, necessitating the development of strategies that either bypass or normalize the BBB to ensure consistent drug delivery throughout the tumor.
6-7) Thank you for your comment. We have addressed the role of IRF8 promoter hypomethylation in contributing to immune evasion in aggressive GBM subtypes in lines 169–173, highlighting how reduced IRF8 expression impairs antigen presentation and facilitates an immunosuppressive microenvironment.
Furthermore, we elaborated on the potential therapeutic role of restoring IRF8 expression in lines 174–177, including its relevance to immune reactivation. While the role of IFN-γ is not the main focus of this section, we acknowledge that IRF8 is a downstream target of IFN-γ signaling, and its modulation may enhance IFN-γ–mediated immune responses, which could be explored in future studies
8) Thank you for your insightful comment. To clarify the significance of the observed correlation between macrophage density in the tumor microenvironment and patient survival in GBM, we expanded our discussion under the section on tumor-associated macrophages (TAMs). Specifically, we referenced prior findings showing that macrophages in gliomas exhibit increased expression of B7-H1, which induces T cell apoptosis and contributes to immunosuppression (PMID: 23613317).
Additionally, we further emphasized this point by noting that "an increased number of macrophages is associated with immunosuppression, which in turn promotes tumor development, invasion, angiogenesis, and resistance to therapy, ultimately leading to poor outcomes" (lines 179–181). This elaboration aims to directly address the clinical significance of macrophage density in GBM.
9-10)
We were unclear about the exact intent of the comment, “What might epigenetic drugs like guadecitabine be used to reverse immune suppression in GBM?” However, to address this, we have elaborated on the mechanism by which epigenetic drugs such as guadecitabine modulate the immune response in GBM. Specifically, we detailed how guadecitabine can reverse immune suppression and promote a more immunogenic tumor phenotype, contributing to therapeutic effects (lines 182–188).
Reviewer 2 Report
Comments and Suggestions for Authors
The manuscript of review “Emerging Chemotherapy Targets: Insights from Advances in 2 Glioma Treatment” highlights recent developments in chemotherapies for treatment of glioblastoma. It outlines challenges faced for treatment of glioblastoma, including blood brain barrier, resistance mechanism, modulation of immune system, and toxicity level and how these new developments potentially overcome these challenges.
Overall, this is a well-written manuscript that looks at important aspect in glioblastoma therapies. It incorporates proper citations and is written in logical manner. Here are some points to improve this manuscript:
1) Provide illustration on how each challenge are addressed by the new advancements.
2) Further discuss how the new advanced in therapies overcome resistance mechanism

Author Response
Thank you for your thorough comments! We have thoroughly addressed the major challenges associated with GBM treatment throughout the manuscript. In particular, Section 4 is dedicated to discussing these therapeutic obstacles and exploring potential strategies to overcome them. Below are selected excerpts from the text that directly address these issues.
Additionally, we expanded the manuscript to include more recent and specific approaches for targeting resistant GBM subtypes. These include strategies such as retroviral gene therapy targeting IRF8 hypomethylation, the therapeutic use of exogenous VEGF, and other novel interventions aimed at improving treatment efficacy and overcoming current limitations.
“To enhance therapeutic delivery, strategies such as intrathecal chemotherapy, focused ultrasound with microbubbles, and nanoparticles capable of crossing the BBB can be employed as summarized in Figure 2”, “Another means of overcoming BBB is by using focused ultrasound and microbubbles to enhance delivery”, “ nanoparticles have shown promising outcomes for the delivery of drugs directly to primary brain tumors through various routes”, “personalized treatment plans for glioblastoma, where genetic profiling informed the use of specific targeted therapies, resulting in improved patient outcomes”, “Another approach to decreasing the incidence of resistance and relapse is by targeting the tumor cell environment, such as stromal cells, immune cells, and other cells that aid in the growth of the tumor. “, “A study reviewing the economic burden of glioblastoma and the cost-effectiveness of pharmacologic treatments, such as temozolomide and carmustine wafers, found that while these treatments may offer better clinical benefits, their high costs present a challenge for cost-effectiveness.”,
Reviewer 3 Report
Comments and Suggestions for Authors
In this review, i couldn't find classification on new therapy either as clinical trials or in vivo studies. Authors mostly focused on chemotherapy while in this area of research, newly approaches focused on targeted gene therapy such as OEC-TK or other type of killing agents. In addition , nanomedicine,either using approved drugs as a nanoformulation or targeting ferroptosis pathway is highly attentive. almost all of these new part missed in this review. i suggest to improve this manuscript by partitionizing it to what we do and what we willing to improve for GBM therapy. The newly approaches for GBM therapy should be extended.
Author Response
Thank you for your insightful comment. To address the increasing research emphasis on modulating the tumor microenvironment and reversing immune evasion, we elaborated on the role of epigenetic modulators—specifically guadecitabine—under the “Immunomodulation” section, describing how these agents promote antigen presentation and enhance immune responses against GBM.
In the section “Emerging Targets in Glioma Chemotherapy,” we incorporated recent developments in chimeric antigen receptor (CAR) T-cell therapy, highlighting its potential to target GBM-specific antigens. We also included a dedicated discussion on nanoparticle-based therapies, emphasizing their dual role in delivering therapeutics across the heterogeneous blood–brain barrier (BBB) and enhancing drug efficacy.
Furthermore, we introduced mitochondrial targets such as PDHA inhibitors, which are currently under evaluation in phase III clinical trials, and discussed their relevance in disrupting tumor metabolism. Finally, we added a section on targeted gene therapy strategies, specifically the use of olfactory ensheathing cells engineered to express HSV-thymidine kinase (OEC-TK), which can selectively alter the tumor microenvironment and induce tumor-specific cytotoxicity.
Round 2
Reviewer 3 Report
Comments and Suggestions for Authors
No further comments